# Prevalence of antibodies against dengue virus in the city of Buenos Aires: Results of a probabilistic population survey

**Cristián Biscayart[1], Patricia Angeleri[1], María Belén Bouzas[2], Lilia Mammana[3], Alejandro Macchia[1]\***

**1** Fundación GESICA (Grupo de Estudios Sobre Investigación Clínica en Argentina), Ciudad Autónoma de Buenos Aires, Argentina, **2** División Análisis Clínicos, Hospital "Francisco J. Muñiz", Ciudad Autónoma de Buenos Aires, Argentina, **3** Unidad de Virología, Hospital "Francisco J. Muñiz", Ciudad Autónoma de Buenos Aires, Argentina

\* alejandro.macchia@fundaciongesica.org.ar

## Abstract

### Objective

Dengue is a rapidly growing global health issue. In Argentina, the disease burden is primarily based on case reports or retrospective series; however, there are no prospective probabilistic studies available. To estimate the seroprevalence of antibodies against the dengue virus (DENV) in adult residents of Buenos Aires, at the end of the 2023 season, which culminated in Argentina's largest recorded dengue outbreak, and immediately before the subsequent outbreak at the beginning of 2024.

### Methods

A stratified, multi-stage probabilistic population survey was conducted to ensure representativeness of the adult population of Buenos Aires, considering geographical and socio-economic diversity. The survey determined the seroprevalence rate of DENV IgG and IgM antibodies using a rapid chromatographic immunoassay. Antibody positive participants were invited for re-testing to confirm antibody result using an enzyme-linked fluorescence assay (ELFA) technique.

### Results

Out of a sampling frame of 2,998 selected households, the interviewers visited 100% of the residences. The overall response rate was 26.8%. A total of 804 participants were enrolled, representing the city's 2.38 million adult population. Weighted estimates showed a total of 193,707 people aged 18 and over have antibodies to DENV. This implies a weighted prevalence [95% CI] of 8.12% [8.08%–8.15%]. The prevalence varied across different areas of the city. In the slums, the overall prevalence was twice as high (14.7% [14.5%–14.9%]) compared to the rest of the city (7.67% [7.63%–7.70%]).

**Data availability statement:** The dataset supporting the findings of this study is publicly available at Zenodo: https://doi.org/10.5281/zenodo.14216600. The dataset is anonymized to protect participant privacy and is provided under a CC BY 4.0 license

**Funding:** This work was supported by an unrestricted grant from Merck (reference 100817 to Fundación GESICA). The funders had no role in study design, data collection and analysis, decision to publish, or preparation of the manuscript.

**Competing interests:** The authors have declared that no competing interests exist

## Conclusion

Seroprevalence results in adults in Buenos Aires show a moderate DENV infection rate, although still far from the thresholds usually considered for initiating a mass vaccination campaign. These results underscore the need for frequent surveys.

## Author summary

Dengue fever is a significant public health threat in many parts of the world, including Latin America. Understanding the seroprevalence of dengue virus antibodies in specific populations helps gauge exposure levels and inform public health decisions. In this study, we conducted a population-based probabilistic survey to estimate the seroprevalence of dengue virus antibodies among adults in Buenos Aires, Argentina, just before a major outbreak in early 2024. We found a moderate level of exposure to the dengue virus, with marked differences across various social and demographic groups within the city. Notably, younger individuals, females, and residents of slums were more likely to have been exposed to the virus. Despite the moderate seroprevalence, our findings indicate that the levels are still below those typically considered necessary for mass vaccination campaigns. These results emphasize the need for regular seroprevalence surveys to monitor the burden of disease and adjust public health strategies accordingly. Our study contributes to a better understanding of dengue epidemiology in urban settings and highlights the importance of localized data in managing and controlling dengue outbreaks.

## Introduction

Dengue (DEN) infection is a global threat, with an estimated half of the world's population at risk of infection [1]. However, the transmission dynamics in various countries remain poorly understood. Understanding these dynamics is crucial for the proper implementation of health policies. In Argentina, dengue virus (DENV) was detected in 1997 after 81 years of epidemiological silence [2]. Since its re-emergence, outbreaks have been recorded frequently across almost the entire country, with all four serotypes identified throughout the period, displaying a heterogeneous spatiotemporal distribution pattern, even within the same jurisdiction. DEN-1 was the most prevalent serotype [3].In 2023, Argentina experienced its largest outbreak in history in terms of cases and geographic extent, with the unprecedented occurrence of autochthonous cases during the winter months in the northeastern provinces and the introduction and widely circulation of DEN-2 serotypes (Cosmopolitan genotype). According to the data available on the Health Information Platform of the Americas, managed by the Pan American Health Organization (PAHO), a total of 841 dengue cases were reported in Argentina throughout 2022. In contrast, 2023 saw a total of 146,876 reported cases [4]. The outbreak in 2023 began around epidemiological week 8 and persisted until approximately week 22 of the same year. After that, no additional cases were reported until the beginning of week 48 in 2023 [5]. Subnational data reported by Argentina's epidemiological bulletins also reflect this significant change in the burden of disease. In Buenos Aires, using reported case data for those aged ≥18 years, the number of affected individuals was 16,138,294 (91% serologically confirmed) as of November 2023, generating an estimated prevalence of approximately 682 cases per 100,000 inhabitants [5].

This outbreak was characterized by the co-circulation of DENV-1 and DENV-2, the latter being predominantly of the Cosmopolitan genotype. The ecology of dengue in Buenos Aires is influenced by urbanization, climate variability, and the widespread presence of *Aedes aegypti*, the primary mosquito vector. High population density and unplanned urban growth, particularly in informal neighborhoods (slums), create ideal conditions for vector proliferation and dengue transmission. Slums, representing approximately 7% of the city's dwellings, are characterized by poor infrastructure, inadequate water and sanitation services, and higher vulnerability to *Aedes aegypti* proliferation. The population living in slums, representing 7% of the city's total population, has distinctive characteristics compared to the rest of the population. Specifically, it is a younger population, with individuals aged 18 or younger accounting for 43% in slums compared to 21% in private housing. The population density is also significantly higher in slums (40,768 vs. 14,555 people per km²). Additionally, poverty and extreme poverty levels are markedly different, with 66.5% of the slum population living below the poverty line compared to 7.4% in private housing. Other notable disparities include the lack of basic infrastructure: 27.1% of households in slums lack proper sanitary services (compared to 0.6% in private housing), 6.4% lack access to piped water (vs. 0.3%), 26.3% lack connection to a sewage network (vs. 0.7%), and 38.4% lack storm drainage (vs. 1%). In contrast, non-slum areas generally benefit from better urban infrastructure. Although *Aedes aegypti* is the predominant vector in Buenos Aires, *Aedes albopictus* has not been documented in the city, likely due to climatic and ecological factors

Dengue surveillance in Buenos Aires relies on a mandatory notification system integrated into the national epidemiological framework. Cases are identified through a combination of clinical criteria and laboratory confirmation, which include PCR, NS1 antigen detection, and serology. Vector control measures implemented by the city's public health authorities include regular fumigation in high-risk areas, education campaigns targeting mosquito breeding sites, and community participation to eliminate standing water. These measures aim to reduce *Aedes aegypti* populations, although their effectiveness is influenced by environmental and socio-economic factors.

The advent of vaccines has increased social pressure to include vaccination in a national plan. However, recommendations for implementing a vaccination program must consider many factors, among which the burden of disease is one of the most important. On the other hand, tetravalent dengue vaccines, such as Sanofi CYD/TDV and Takeda's TAK-003, have shown safety and efficacy concerns, especially in seronegative individuals [6]. Recent studies have indicated a negative efficacy in preventing hospitalizations for dengue in these individuals, highlighting the need for more thorough research and a cautious approach to the implementation of these vaccines to avoid the risk of vaccine-induced severe infections [6].

Considering the heterogeneity in the distribution of dengue disease both between and within countries, the WHO Strategic Advisory Group of Experts (SAGE) recommends the implementation of local epidemiology plans [7]. In this manuscript, we report the first determination of dengue seroprevalence in Argentina, particularly in the City of Buenos Aires, using a probabilistic approach.

## Methods

### Ethics statement

The investigation protocol was reviewed and approved by human subjects review experts from the Institutional Review Board (IRB) at the Comité de Ética Centro de Osteopatías Médicas (CECOM). Informed written consent for survey participation and blood collection was obtained from all participants. The study was registered on ClinicalTrials.gov with the following identifier: NCT05919277.

## Study Area

Buenos Aires is the largest city in Argentina and has a population of 3.1 million of inhabitants, with 2.38 million having 18 or more years of age.

## Study design

A stratified, multi-stage probabilistic population survey was conducted among adults (18 years and older) residing in Buenos Aires to estimate the seroprevalence of DEN (presence of Dengue IgG and/or Dengue IGM antibodies) among the city's inhabitants. Information was collected by visiting only private households, selected through a probabilistic sampling method. Consequently, the sample results were valid representations of the entire city. This approach allowed for the inclusion of diverse subpopulations by dividing the sampling frame into strata based on relevant criteria, such as income levels and geographic areas. By employing multi-stage sampling, we optimized resource allocation while maintaining statistical reliability, as each stage progressively narrowed the selection from broader units (e.g., census tracts) to individual households and participants. This methodology reduced sampling bias and enhanced the validity of results for an urban area as complex and diverse as Buenos Aires.

The fieldwork and surveys were carried out in collaboration with the General Directorate of Statistics and Censuses of Buenos Aires, using the two current sampling frames for household surveys. The first frame, referred to as the general private households' frame, comprises approximately 93% of the dwellings in Buenos Aires, while the second frame, called the "Informal Popular Neighbourhoods" frame, includes the dwellings in the city's slums. The primary sampling areas (PU) were mostly defined as the 'census tracts' from the 2010 national population, household, and housing census, considering that census tracts/PU areas consist of a set of 350 contiguous dwellings. These PU areas were classified into five strata, constructed through a functional relationship between income quintiles obtained from the annual household survey and variables collected in the census. Stratum 1 represents the lowest-income population, while Stratum 5 corresponds to the highest-income population.

The first sampling stage of this frame involved selecting 300 PU areas. These were selected with a probability proportional to size, with the measure of size being the total number of dwellings. In the second stage, 10 dwellings were selected in each of the 300 PU areas, resulting in a sample size of 3,000 dwellings for this frame. The selected dwellings were visited by interviewers and nurses, who randomly selected one person to complete the survey. This final selection constituted the third and last stage of sampling. Within the context of household sampling, the respondent in the household was randomly selected using the Kish method, which is widely used in field research to ensure a random and equitable selection of participants within a household. For the 'Informal Popular Neighbourhoods' sampling frame, census tracts from slum areas in Communes 1, 4, 7, 8, 9, and 15 were used as Primary Sampling Units (PSUs). A three-stage sampling design was applied: 20 PSUs were selected with probability proportional to size, 10 dwellings were randomly chosen within each PSU, and one resident was randomly selected per dwelling. This approach ensured representativeness of the slum population within the study. These areas are distinct in their socio-economic conditions and environmental risks, which influence mosquito breeding and dengue transmission dynamics. Interviews were conducted in-person by trained professionals, following a structured protocol to ensure consistency and accuracy. All adults over 18 years of age residing in selected households were eligible to participate, with exclusions as described in the eligibility criteria (e.g., individuals vaccinated against yellow fever or dengue).

To be eligible for this study, participants had to be adults aged 18 or older, residents of Buenos Aires, and provide informed consent. The exclusion criteria included those unable

to provide consent and individuals who reported having been vaccinated against yellow fever or dengue.

As with any complex sample, there is an inherent risk of bias. Consequently, it is common practice to calibrate the initial weights or expansion factors using auxiliary information that is known or pre-established from records or external sources. For the adjustment of the initial weights, the technique of 'calibration by fixed margins' was used, following the methodology developed by Deville and Särndal [8]. This process used auxiliary data from DGESYC population projections and the 2022 Annual Household Survey, ensuring alignment of the weighted sample estimates with known population totals by sex and age groups. Calibration was performed independently for each of the three sampling zones and included all strata defined in the survey design.

The sample size estimation assumed an anticipated seroprevalence of 10%, based on historical data and prior studies of dengue exposure in urban populations with similar characteristics. The sample size was calculated to achieve a power of 90% with a 5% margin of error and a design effect of 1.5 to account for the complex sampling design. Based on these parameters, the required sample size was estimated at 1,487 households. This number was subsequently adjusted to 3,000 dwellings to account for potential non-response.

All estimates and comparisons were made using the expanded value. Continuous variables were compared using either the t-test or the Mann-Whitney test, as appropriate..To identify factors associated with dengue seropositivity, multivariable logistic regression models were applied, given that the outcome variable (seropositivity) is binary. Independent variables included sex, age groups, household size, living in slum areas, and a history of dengue within the household. The results are presented as odds ratios (ORs) with 95% confidence intervals (CIs).

## Procedures

Each household was visited by a professional interviewer and a healthcare professional (registered nurse). Participants were informed about the study's objectives and scope and proceeded to give their consent by signing an informed consent form. The interviews were conducted using a structured questionnaire designed to collect demographic information, medical history, and knowledge of dengue. The questionnaire was pre-tested in a pilot study to ensure clarity and reliability. The nurses performed a finger-prick to conduct a rapid test that detected the presence of Dengue IgG antibody and Dengue IgM antibodies using a rapid immunochromatographic assay (Laboratorios JAYOR SRL-Argentina). The samples tested consisted of capillary blood obtained through a finger-prick. Blood was directly applied to the test strip of a rapid immunochromatographic assay, following the manufacturer's instructions. The assay does not require additional dilution of the blood sample, as it is designed to analyze whole blood directly. The performance parameters of this test, as reported by the manufacturer, indicated a sensitivity of 95.8%, 95% CI: 91.1–98.4%, and a specificity of 96.1%, 95% CI: 92.6–98.4%, compared to ELISA assays. For individuals with positive IgG results in the rapid test, venous blood samples were collected and processed at the Virology Unit of the Hospital F.J. Muñiz. Serum Dengue IgG antibody assay (bioMérieux-France), which employs the ELFA (enzyme-linked fluorescence assay) technique. The assay was performed following the manufacturer's instructions, which include an automatic dilution step by the VIDAS system. This involves mixing the serum sample with reagents and applying it to a solid-phase device coated with recombinant dengue antigens. The fluorescence intensity measured at the end of the process is proportional to the concentration of IgG antibodies present in the sample. No manual dilution of the serum was performed. This system includes a predefined automatic dilution of 1:400 within the analytical process, guaranteeing standardization and reproducibility of the results. Controls for the rapid test included manufacturer-provided positive and negative

samples, which were run during each testing session. The VIDAS Dengue IgG antibody assay was selected for confirmation due to its higher specificity (99.1%) and semi-quantitative design, which complements the qualitative nature of the rapid test.

## Results

### Sampling and response rate

Participant recruitment was conducted between epidemiological weeks 34 and 48 of 2023, during a period of no active dengue transmission in Buenos Aires, according to official data from the National Epidemiological Bulletin and PAHO.

Out of the 2,998 selected households, 804 individuals participated in the study, resulting in a response rate of 26.8%. Weighted estimates were used to adjust for non-response and to provide prevalence rates representative of the total adult population of Buenos Aires (2.38 million). The reasons for non-response included the absence of residents at the household despite three separate visits on different days (30.2%), refusal to participate in the study (37.8%), unoccupied dwelling (18.8%), inability to participate due to vaccination against yellow fever (7.5%), refusal of the individual to provide a sample after signing the consent form (4.7%), and other reasons (1.1%)..

### Seroprevalence

A total of 804 participants were tested for DENV antibodies. Based on these results, 71 individuals tested positive, corresponding to a weighted prevalence of 8.12% [95% CI: 8.08%–8.15%], equivalent to approximately 193,707 adults in the city's population. Of the 71 participants who tested positive by the rapid test, 27 underwent confirmatory testing with the ELFA assay, all of which were confirmed as positive (100% concordance). Among the population, 6.3% had IgG antibodies, 1.5% had IgM antibodies, and 0.3% had both types. A higher prevalence [95% CI] was observed in females, 9.35% [9.29%–9.40%], compared to males, 6.69% [6.6%–6.8%] (p<0.0001). Fig 1 shows the number of individuals who had a positive antibody response by age groups.

Positive individuals had a mean age of 41.8 (17.4) years, while negatives had a mean age of 47.5 (17.6) years (p<0.0001). The main characteristics of the population according to the results of the seroprevalence study are summarized in Table 1. Participants who tested positive were more frequently in the 18 to 34 age group compared to those who tested negative. However, the comparison of mean and median age was at the threshold of statistical significance. Similarly, those who tested positive were more often residents of the city's slums and more frequently reported having experienced clinically evident dengue. The remaining demographic variables, personal history, previous symptoms, and comorbidities were similar between participants who tested positive and those who tested negative for antibodies (Table 1). Four factors were independent predictors of testing positive for antibodies: a self-reported history of dengue (9.13 [9.03–9.24]), living in a slum (1.48 [1.45–1.50]), being female (1.58 [1.56–1.60]), and belonging to the 18 to 34 age group (3.25 [3.20–3.30]).

There was a heterogeneous distribution of prevalence across Buenos Aires, with the highest concentrations in the central region of the city (10,9%) followed by southern communes (6.3%) and northern communes (4.42%).

The factors associated with dengue seropositivity are shown in Table 2. Females were more likely to test positive compared to males (OR = 1.442, 95% CI: 1.429–1.456). Living in a slum area was associated with higher seropositivity (OR = 1.151, 95% CI: 1.133–1.169), as was an increasing number of household members (OR = 1.039, 95% CI: 1.036–1.043). Age was

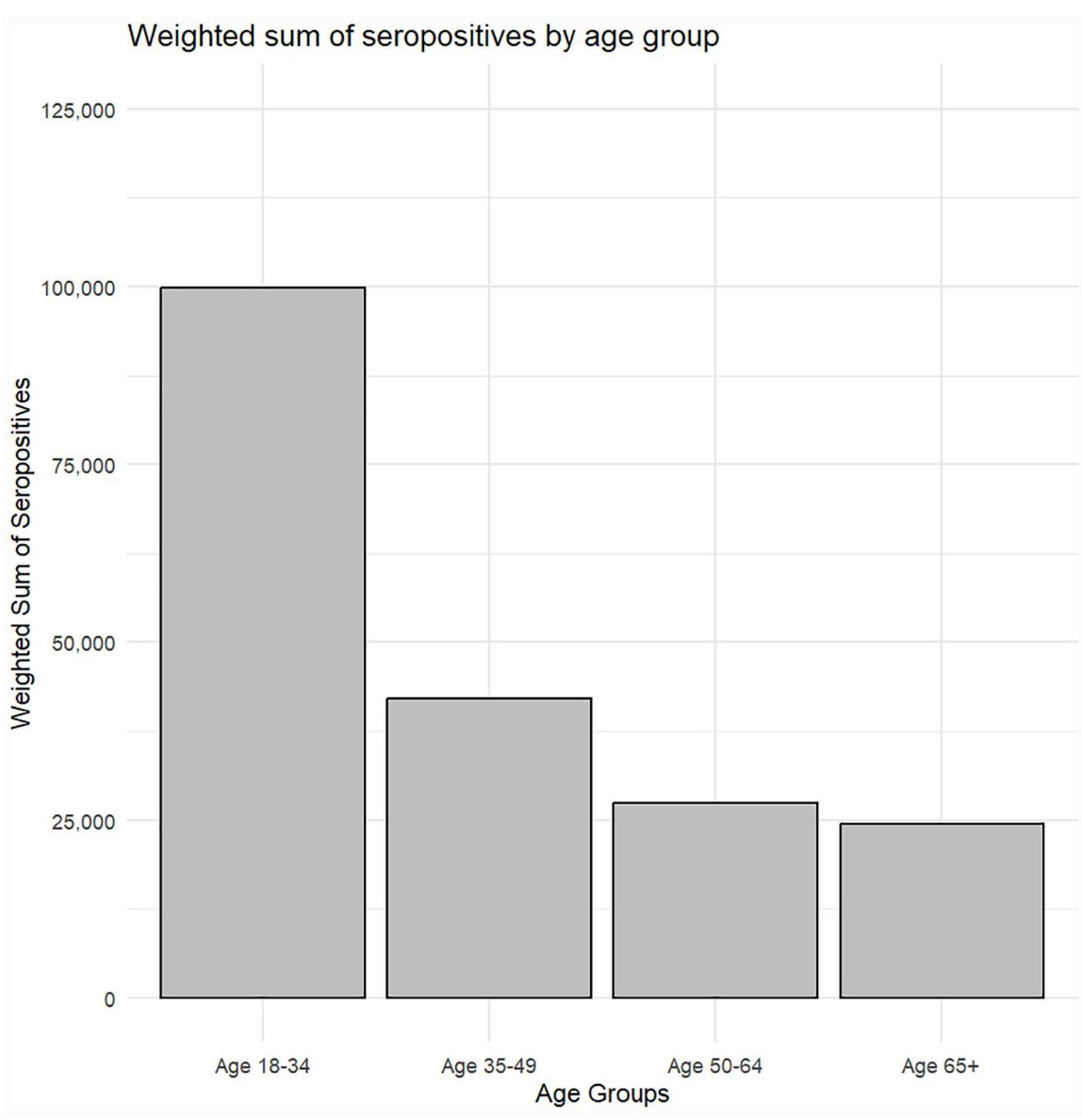

**Fig 1. Number of individuals who had a positive antibody response by age groups.**

inversely associated with seropositivity, with individuals aged 18–35 years being significantly more likely to test positive (OR = 2.604, 95% CI: 2.567–2.641) compared to those aged 65 years and older (reference group). Finally, having a household history of dengue was strongly associated with seropositivity (OR = 5.762, 95% CI: 5.709–5.817).

## Discussion

This work reports for the first time, using a probabilistic estimation, the seroprevalence of dengue in Argentina, particularly in the City of Buenos Aires. The overall seroprevalence indicates a low level of endemicity for dengue in Buenos Aires, consistent with its history of recent reintroduction and episodic outbreaks. While this rate suggests a lower force of infection compared to hyperendemic settings, it remains significant for public health

**Table 1. Characteristics of participants according to the antibody test result.**

| | Antibody test result | | | | | | p-value |
|---|---|---|---|---|---|---|---|
| | Negative (crude) | Negative (weighted) | % | Positive (crude) | Positive (weighted) | % | |
| n | 733 | 2, 192,681 | 100 | 71 | 193,707 | 100 | |
| **Demographic Characteristics** | | | | | | | |
| **Age groups** | | | | | | | |
| 18–34 years | 144 | 595,269 | 27.1 | 23 | 99,896 | 51.6 | 0.01855 |
| 35–49 years | 220 | 645,550 | 29.4 | 21 | 42,026 | 21.7 | |
| 50–64 years | 196 | 460,945 | 21 | 16 | 27,324 | 14.1 | |
| 65+years | 173 | 490,917 | 22.4 | 11 | 24,461 | 12.6 | |
| Age (mean, sd) | | 47.5 (17.6) | | | 41.8 (17.4) | | 0.0627 |
| Age (median, p25%,p75%) | | 46 (33-63) | | | 34 (28-53) | | 0.057 |
| Female sex (n, %) | 421 | 1, 163,123 | 53.0 | 46 | 119,914 | 61.9 | 0.00001 |
| **Type of Household** | | | | | | | |
| Slums (n, %) | 171 | 127,830 | 5.8 | 30 | 22,096 | 11.4 | 0.00001 |
| Appartment/house (n, %) | 522 | 1, 968,248 | 89.8 | 39 | 164,530 | 84.9 | |
| Social house (n, %) | 40 | 96,603 | 4.4 | 2 | 7,081 | 3.7 | |
| Number of residents in the household (n, %) | | 2.5 (1.4) | | | 2.92 (1.9) | | 0.292 |
| **Background and Knowledge** | | | | | | | |
| Self-reported prior dengue (n, %) | 63 | 68,726 | 3.1 | 27 | 76,848 | 39.7 | 0.00001 |
| Dengue Information (*) (n, %) | 587 | 1, 823,687 | 85.9 | 39 | 91,525 | 78.3 | 0.496 |
| Believes they could get dengue again in the future (*) (n, %) | 16 | 42,569 | 61.9 | 16 | 47,275 | 61.5 | 0.984 |
| Believes dengue can be severe (n, %) | 564 | 1, 672,334 | 76.3 | 61 | 162,256 | 83.8 | 0.853 |
| **Symptoms** | | | | | | | |
| Fever (n, %) | 104 | 329,374 | 15 | 12 | 47,120 | 24.3 | 0.243 |
| Chills (n, %) | 114 | 335,236 | 15.3 | 14 | 44,984 | 23.2 | 0.32 |
| Fatigue (n, %) | 215 | 618,272 | 28.2 | 21 | 65,202 | 33.7 | 0.559 |
| Retro-ocular pain (n, %) | 66 | 186,002 | 8.5 | 11 | 20,870 | 10.8 | 0.586 |
| Headache (n, %) | 165 | 523,257 | 23.9 | 20 | 52,191 | 26.9 | 0.735 |
| Myalgia/Arthralgia (n, %) | 205 | 571,932 | 26.1 | 22 | 68,395 | 35.3 | 0.306 |
| Mucosal Bleeding (n, %) | 49 | 175.062 | 8,0 | 4 | 9.699 | 5,0 | 0,451 |
| **Comorbidities** | | | | | | | |
| Hypertension (n, %) | 198 | 496.275 | 22,6 | 13 | 31.424 | 16,2 | 0,345 |
| Diabetes (n, %) | 67 | 116.317 | 5,3 | 6 | 6.728 | 3,5 | 0,522 |
| Cardiovascular history (n, %) | 60 | 159.004 | 7,3 | 5 | 8.518 | 4,4 | 0,389 |
| Rheumatologic history (n, %) | 63 | 133.018 | 6,1 | 7 | 17.244 | 8,9 | 0,499 |
| **Anthropometry** | | | | | | | |
| Weight (kg) (mean, sd) | | 74,9 (7,17) | | | 75,7 (6,19) | | 0,851 |
| Height (cm) (mean, sd) | | 166,8 (4,10) | | | 166,2 (3,9) | | 0,729 |

*The denominator for participants without prior dengue is 2,123,955 for the negatives and 116,859 for the positives. The denominator for participants with prior dengue was 68,726 for the negatives and 76,848 for the positives.

planning given the city's vulnerability to future outbreaks and the potential for increasing transmission intensity. Seroprevalence studies are a important tool in epidemiological surveillance, as they allow for an assessment of the population's overall exposure to the virus, including asymptomatic infections that are not captured by case notification systems. However, seroprevalence alone is not sufficient to determine disease burden or guide public

Table 2.  Factors Associated with Seropositivity.

|  | OR | Lower 95% CI | Upper 95% CI |
|---|---|---|---|
| Female sex | 1.442 | 1.429 | 1.456 |
| Living in a slum area | 1.151 | 1.133 | 1.169 |
| Number of household residents | 1.039 | 1.036 | 1.043 |
| Age ≥65 years | 1 | | |
| Age ≥50 y <65 years | 0.91 | 0.894 | 0.926 |
| Age ≥35 y <50 years | 0.875 | 0.861 | 0.889 |
| Age ≥18 y <35 years | 2.604 | 2.567 | 2.641 |
| History of dengue in the household | 5.762 | 5.709 | 5.817 |

health policy. Its interpretation must be complemented with traditional epidemiological surveillance data and mathematical models that can estimate the true incidence of the disease and assess the impact of public health interventions [9,10]. So far, the epidemiological description of dengue in Argentina has been limited to cases reported by the national epidemiological surveillance system. Although dengue is a notifiable disease, it is likely that the number of reported cases underestimates the disease burden due to the presence of asymptomatic or mildly symptomatic cases, as well as a tendency for underreporting communicable diseases. In Buenos Aires, using reported case data for those aged ≥18 years, the number of affected individuals was 16,138 (91% serologically confirmed) as of November 2023, generating an estimated prevalence of approximately 682 cases per 100,000 inhabitants. In contrast, our study suggested a disease burden of 9,050 cases per 100,000 inhabitants. The disparity between reported cases and disease burden as measured by seroprevalence indicated that many dengue infections are asymptomatic, and the cases reported through surveillance systems represented only a small fraction of the actual cases [10]. Indeed, studies highlighting these disparities also estimate that the results derived from data collected using faster and cheaper diagnostic tests were comparable to those obtained from more expensive tests, an important conclusion for surveillance in resource-limited countries [10]. These findings reinforce the need to establish a robust system for assessing disease burden through serial population-based seroprevalence studies and to discount—at least for making significant health decisions—those based on reported cases [11], nor in studies that do not use a probabilistic approach, such as the use of samples from blood banks [12]. However, it should also be noted that while seroprevalence cannot directly measure the clinical burden of dengue, it serves as a complementary tool to active and passive surveillance. By identifying immunity gaps and historical exposure, serial seroprevalence studies provide baseline data critical for modeling future outbreaks, guiding vaccination strategies, and evaluating vector control measures. However, their use should always be contextualized within a broader surveillance framework

Seroprevalence was heterogeneous across different regions and social strata within the City of Buenos Aires. Our study found a higher rate of antibodies among those individuals who had a household member with a self-reported history of the disease. Additionally, younger age groups and residents of slums were more likely to test positive for antibodies. The age distribution of DENV antibody-positive participants showed a higher prevalence in younger adults compared to older age groups. This pattern contrasts with hyperendemic areas, where cumulative lifetime exposure leads to higher seroprevalence in older individuals. However, other estimates from the Global Burden of Disease Study indicate that the highest incidence of dengue occurs among adolescents and young adults [13]. Factors such as urban mobility,

social interactions, and the recent expansion of outbreaks in high-density areas may contribute to this trend. This observation highlights the importance of targeted prevention strategies and continuous epidemiological surveillance to monitor potential shifts in transmission dynamics. The higher seroprevalence observed in slums compared to non-slum areas is consistent with the ecological and socio-economic disparities described in the literature. While *Aedes albopictus* is not currently present in Buenos Aires, continuous monitoring is essential given the potential for environmental changes to alter vector distribution. These findings were consistent with those reported in the literature [14,15].

These results highlight important associations with dengue seropositivity, such as higher rates among younger adults, females, individuals living in slum areas, and those with household members previously infected by dengue. However, it is important to note that this study was designed primarily to describe seroprevalence, and these associations should be interpreted cautiously in the context of this descriptive study.

Although our work is a population survey aimed at the epidemiological description of dengue burden among adults residing in the City of Buenos Aires, it is inevitable to comment on how these findings relate to the need for recommending a vaccination strategy. Historically, DEN-1 was the predominant circulating serotype in Buenos Aires City, but since 2023, DEN-2 has started circulating. In fact, in the past and current seasons infections in the City of Buenos Aires were primarily due to serotype 2 and, to a lesser extent, serotype 1. According to data from the latest epidemiological bulletin, 68% of cases were from DEN-2, while 32% are from DEN-1, suggesting that among seropositive citizens there is homotypic immunity to both viruses, especially for DEN-2.

The pivotal study (TIDES) of TAK-003 vaccine [16] modest efficacy was reported for serotype 1 (43.5%) and high efficacy for serotype 2 (92%), with no efficacy for serotypes 3 (-23%) and 4 (-105%) during the first 36 months of the trial. In initially seronegative persons, clinical data for TAK-003 show no efficacy against serotypes 3 and 4 at any time, and no efficacy against serotype 1 after 24 months. The data also indicate a trend toward increased risk of serotype 3 disease and hospitalization in seronegative vaccinated children in Sri Lanka, although this trend was not statistically significant. Some experts believe that TAK-003 is not exempted from the possibility of triggering the ADE phenomenon in seronegative vaccinees. All these results would suggest the use of this vaccine especially in people with a history of previous infection. Extensive preclinical and postclinical research on TAK-003 indicates that its immunogenicity is primarily driven by serotype 2 component, with considerably less or no contribution from the other three vaccine components [16,17]. TAK-003 is the only vaccine approved by the national regulatory agency in Argentina.

In a recent statement, the WHO SAGE on immunization [18] asserted that there is no precise age-specific seroprevalence threshold above which vaccination is indicated. Rather, the benefit of vaccination increases with higher seroprevalence, with better vaccine performance expected in seropositive individuals. The decision on the threshold cut-offs for minimal seroprevalence to initiate vaccination should be made at the country level. Typically, a seroprevalence at the age of 9 years old of over 60% could be considered an indicator of high dengue transmission. Conversely, the PAHO Technical Advisory Group of Experts on Immunization has decided not to recommend the implementation of nationwide immunization programs with the TAK-003 vaccine at this time. Furthermore, some countries in the Region of the Americas may wish to introduce the TAK-003 vaccine in specific subnational geographic areas where there is documented evidence of a high burden of dengue and high transmission intensity [19]. Our findings for the city of Buenos Aires do not, to date, appear to fall into this epidemiological risk category. The application of the Medical Research Council (MRC) equations (7) to the population of the City of Buenos Aires indicates a low force of infection

and a significant number of years required to reach the current recommendation threshold. Consequently, there is currently no evidence to support the use of this vaccine in seronegative residents of the City of Buenos Aires as a public health policy.

. Our study has strengths and limitations that should be mentioned. Firstly, participant recruitment had to be stopped earlier than planned due to the emergence of a new dengue epidemic starting in epidemiological week 48 of 2023. Since the study aimed to estimate pre-epidemic seroprevalence, continuing recruitment during the outbreak would have resulted in a dataset reflecting both post-epidemic immunity from 2023 and new infections from the ongoing epidemic, making interpretation difficult.

Considering that recruitment began in September 2023—a period with no reported dengue cases for several months—the collected data accurately represent the seroprevalence status before the new epidemic. The low IgM prevalence (1.5%) is consistent with the period in which the study was conducted, when viral circulation was still low, aligning with the epidemiological dynamics recorded in the city throughout 2023. Ongoing surveys will provide information on how this situation may have changed. These ongoing studies are not part of the present study but rather national and subnational initiatives, and they have no relation to this report. Additionally, the non-response rate was higher than expected. The low participation rate introduces the possibility ofse bias. Although calibration by fixed margins was applied to mitigate this issue, the lack of data on non-participants limits our ability to directly assess differences between participants and those who refused to participate. The proportion of participants who underwent confirmatory testing was limited (27/71), which may affect the generalizability of the confirmatory results. However, the 100% concordance observed supports the reliability of the rapid test in this context.

This survey focused on adults (aged 18 years and older) due to logistical and regulatory constraints. In Argentina, including children in research studies presents significant challenges, as it requires obtaining parental consent and navigating complex ethical and regulatory processes. Additionally, practical considerations, such as coordinating with schools or other institutions, would have made the study substantially more difficult to implement within the available timeframe and resources. For these reasons, the study was limited to the adult population, while acknowledging that the findings may have limited generalizability to younger age groups. Moreover, the results of this study cannot be automatically extended to the entire population of Buenos Aires but only to individuals aged 18 years and older. This also poses the challenge that vaccination recommendations are typically based on seroprevalence levels among those aged 5 to 18 years. Although seroprevalence in children tends to be higher than in adults, the likelihood that seroprevalence in children from Buenos Aires would fall within recommendation thresholds is low.

Seroprevalence surveys provide valuable insights into the cumulative exposure of a population to dengue virus (DENV), but they have inherent limitations. Most DENV infections are asymptomatic, and antibody detection reflects prior infections rather than current infection rates. This means that seroprevalence data cannot distinguish between recent and past infections, potentially obscuring short-term epidemiological trends. Despite this, serosurveys remain an essential tool for public health planning, as they help monitor the long-term burden of disease, identify high-risk populations, and evaluate the impact of vector control and vaccination programs. In Buenos Aires, these data complement surveillance systems, which are focused on detecting symptomatic cases, and vector control measures aimed at reducing *Aedes aegypti* populations.

It is important to recognize that dengue transmission typically occurs in spatial clusters, a phenomenon that was not fully addressed in this study. While we used a stratified sampling design and applied weights to reflect the population distribution, these strategies do not fully

capture the spatial dynamics of transmission. Additionally, the limited number of samples per age group and the low seroprevalence values might have reduced the representativeness of the results. Future studies should consider the use of geospatial approaches to better model these dynamics and ensure a more accurate representation of dengue transmission

In conclusion, for the first time, a probabilistic and population-based estimate is reported from Argentina, showing a modest burden of dengue in the adult population of Buenos Aires immediately before the 2024 outbreak. The results necessitate the design and administration of serial population seroprevalence studies to inform health decisions.

## Author contributions

**Conceptualization:** Cristián Biscayart, Patricia Angeleri, Alejandro Macchia.

**Data curation:** Alejandro Macchia.

**Formal analysis:** Alejandro Macchia.

**Funding acquisition:** Cristián Biscayart, Patricia Angeleri.

**Investigation:** Cristián Biscayart, Patricia Angeleri.

**Methodology:** María Belén Bouzas, Lilia Mammana, Alejandro Macchia.

**Project administration:** Patricia Angeleri.

**Resources:** Cristián Biscayart, Patricia Angeleri.

**Supervision:** Cristián Biscayart, Alejandro Macchia.

**Validation:** Patricia Angeleri.

**Writing – original draft:** Alejandro Macchia.

**Writing – review & editing:** Cristián Biscayart, Patricia Angeleri, Alejandro Macchia.

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
