## [Decision Letter · Decision Letter 0]

11 Nov 2024

PNTD-D-24-01249Prevalence of Antibodies Against Dengue Virus in the City of Buenos Aires: Results of a Probabilistic Population Survey.PLOS Neglected Tropical Diseases Dear Dr. Macchia, Thank you for submitting your manuscript to PLOS Neglected Tropical Diseases. After careful consideration, we feel that it has merit but does not fully meet PLOS Neglected Tropical Diseases's publication criteria as it currently stands. Therefore, we invite you to submit a revised version of the manuscript that addresses the points raised during the review process. Please submit your revised manuscript within 60 days Jan 10 2025 11:59PM. If you will need more time than this to complete your revisions, please reply to this message or contact the journal office at plosntds@plos.org. Please include the following items when submitting your revised manuscript:* A rebuttal letter that responds to each point raised by the editor and reviewer(s). You should upload this letter as a separate file labeled 'Response to Reviewers '. This file does not need to include responses to any formatting updates and technical items listed in the 'Journal Requirements' section below.* A marked-up copy of your manuscript that highlights changes made to the original version. You should upload this as a separate file labeled 'Revised Manuscript with Track Changes '.* An unmarked version of your revised paper without tracked changes. You should upload this as a separate file labeled 'Manuscript '. If you would like to make changes to your financial disclosure, competing interests statement, or data availability statement, please make these updates within the submission form at the time of resubmission. Guidelines for resubmitting your figure files are available below the reviewer comments at the end of this letter. We look forward to receiving your revised manuscript. Kind regards, Husain PoonawalaAcademic EditorPLOS Neglected Tropical Diseases David SafronetzSection EditorPLOS Neglected Tropical Diseases

Shaden Kamhawi

co-Editor-in-Chief

Paul Brindley

co-Editor-in-Chief

 **Journal Requirements:** **Additional Editor Comments (if provided):****Reviewers' Comments:** Reviewer's Responses to Questions

**Key Review Criteria Required for Acceptance?**

**Methods**

-Are the objectives of the study clearly articulated with a clear testable hypothesis stated?

-Is the study design appropriate to address the stated objectives?

-Is the population clearly described and appropriate for the hypothesis being tested?

-Is the sample size sufficient to ensure adequate power to address the hypothesis being tested?

-Were correct statistical analysis used to support conclusions?

-Are there concerns about ethical or regulatory requirements being met?

Reviewer #1: See Summary and General Comments

Reviewer #2: (No Response)

**Results**

-Does the analysis presented match the analysis plan?

-Are the results clearly and completely presented?

-Are the figures (Tables, Images) of sufficient quality for clarity?

Reviewer #1: See Summary and General Comments

Reviewer #2: (No Response)

**Conclusions**

-Are the conclusions supported by the data presented?

-Are the limitations of analysis clearly described?

-Do the authors discuss how these data can be helpful to advance our understanding of the topic under study?

-Is public health relevance addressed?

Reviewer #1: See Summary and General Comments

Reviewer #2: (No Response)

**Editorial and Data Presentation Modifications?**

Reviewer #1: See Summary and General Comments

Reviewer #2: (No Response)

**Summary and General Comments**

Reviewer #1: The authors describe their study to determine the prevalence of dengue virus (DENV) antibody in adult residents after the 2023 season and before the beginning of a dengue outbreak in 2024 in Buenos Aires Argentina. A survey was conducted among adults 18 years and older who resided in the city of Buenos Aires with an adult population of 2.38 million people. Blood samples were tested for DENV IgG and IgM antibodies using a rapid chromatographic immunoassay. If positive for antibody, the results were confirmed using an enzyme-linked fluorescence assay (ELFA). Out of a sampling frame of 2,998 selected households, the interviewers visited 100% of the residences. The overall response rate was 26.8%. Weighted estimates indicated that a total of 193,707 participants had DENV antibodies. implying a weighted antibody prevalence of 8.12%. The prevalence rate varied with participants in the slums having a rate twice as high (14.7%) as those in the rest of the city (7.67%). The findings indicated that adults in Buenos Aires had experienced a moderate amount of DENV infections, but according to the authors the amount was under the thresholds usually considered for implementing a mass vaccination campaign and the results underscored the need for frequent surveys.

The rationale for this study was justified by addressing DENV as a global public health concern that is the most frequent cause of human disease of all the mosquito-borne viruses and globally causes an estimated 100–400 million infections, including severe and fatal cases each year. Therefore, the contents of this manuscript are highly appropriate for publication consideration in this journal. However, the suggested comments should be considered for improving the contents of the manuscript. The objective in the abstract should include a brief description of the burden of dengue disease in the population for justifying the decision to perform the survey. The methods in the abstract state that a stratified, multi-stage probabilistic population survey was conducted that should be described for the readers who do not understand and the reasons for selecting this survey method, may be best to describe in the method section. Why were only adults selected for inclusion in this survey, again can be done in the method section? Interviews are mentioned in the results section of the abstract but are not mentioned in the methods, again can be done in the method section as well as the sampling frame should be defined as well as households, did this include all adults over 18 years of age (2.38 million) except those excluded by the eligibility criteria. The methods in the abstract, or in the method section do not state how many participants were actually enrolled in the survey other than stating that the survey was conducted among adults (selected areas and households) residing in the city of Buenos Aires, representing the city’s 2.38 million adult population. An actual number of the participants would be more informative. While the antibody assay techniques are stated, nothing is mentioned as to what samples were tested, assume paper adsorbed blood samples for antibody or how the techniques were performed, assume in accordance to the manufacturer’s instructions and what dilution(s) of the sample were tested, again may be done in the method section. The content of the introduction section could be improved by a brief description of the global dengue disease burden as well as the burden of the disease that outbreaks have caused in Buenos Aires. Also, the contents of the introduction section is focused on dengue vaccines with nothing much included on history of dengue in Buenos Aires, including the ecology and epidemiology of dengue in Buenos Aires and other communities in Argentina. While the rationale for the study is to assess the DENV infection rates for making a decision on a mass vaccination campaign, a more relevant approach would be to base the assessment on the burden of dengue disease. Also, if surveillance for dengue cases is being done, this should be described and what is being done to prevent dengue cases, such as vector control. As the author’s knows, the majority of DENV infections are asymptomatic and therefore, is a weakness of serosurveys for public health purposes as antibody prevalence rates only indicate prior infections and nothing about disease, and also is limited for assessing current infection rate because antibody can represent infections that occurred many years ago. However, the information is of public health value for monitoring the status of the potential burden of disease and for estimating the efficacy of vector and/or vaccine control measures. Some comments on the method section were included with the comments on the contents of the abstract. Additional comments to improve the manuscript pertain to the study area that needs to be describe especially the slums versus the non-slum areas of the city to include any relevant information on Aedes aegypti and/or Aedes albopictus. The use of sampling areas and household as a description of the number of participants enrolled in the study as alluded to above is confusing and could be clarified by including the total number of participants enrolled and tested in the study. It is stated that the selected dwellings were visited by interviewers and nurses, who randomly selected one person to complete the survey. What was the instrument used to perform the interviews and who was the one person selected to participate in the interviews? While the sampling procedure is described, it is not clear as to how reliable the estimated prevalence rate was for the entire city. If the assumption was made that the rate based on a sample of the population would be applicable to the total population, this would need to account for the well established understanding that DENV infection is not randomly distributed but occurs in clusters based on the risk associated with the present and movement of people and places where contact with infected mosquitoes are likely that causes marked heterogeneity in transmission rates in a community. The methods used to perform the assay to test samples for antibody needs to be described in more detail, for examples, in addition to the comments mentioned in the abstract, were controls run in the tests, including antibody positive and negative samples to validate the performance of the test. Although the test according to the manufacturer has a very high sensitivity and specificity, confirmation of these test parameters should have been done prior to use in this study. A second test was mentioned for use to confirm the results of the first test, why did the results of the first test need to be confirmed and what was different about the second test that supported it use as a confirmation assay. Also, the very low samples available from only 27 participants for confirmation by a second test present some concerns regarding the validity of the results of this study. Overall, the results are adequately described, but could be improved by more detailed description to clarify some of the observations. Only 26.8% of 2,998 selected households were interviewed, did this low rate provide data representative of the total study population and was the 26.8% included as participants of this study. The number of participant tested is stated as a total 193,707 people aged 18 and over tested positive for DENV antibodies. To clarify the reviewer’s previous comment on participants in the study, how many participants were enrolled, how many were tested and how many were positive of the total participants tested. The results presented in Table 1 need to be described more-so in the results and discussion sections. For example, the age distribution of antibody positive participants decreased from 51.6% for the 18-34 ages to 25.5% for the 65 and over ages. In a dengue endemic community, like Buenos Aires, especially being a hyperendemic community, the antibody prevalent rate would be expected to be the reverse indicating that the individuals who lived the longest in the community would have the highest antibody prevalent rate. Also, the overall detection of DENV antibody in about 8% of the population is not considered to be a moderate prevalent rate, but a rate that indicate a low risk for DENV infection. For example, seroprevalence rates range from 60 to 80% in other dengue hyper-endemic communities. The discussion section tends to focus on dengue vaccines and should address more-so the observations of this study, for example, it is stated that “Although dengue is a notifiable disease, it is likely that the number of reported cases underestimates the disease burden due to the presence of asymptomatic or mildly symptomatic cases, as well as a tendency for underreporting communicable diseases” Not clear as to the meaning of the latter sentence, as asymptomatic cases would not contribute to the burden of disease, under-reporting would lessen the estimated burden of disease, but is there evidence for under-reporting of cases. In regard to the burden of disease associated with dengue, estimates discussed are based on ≥18 years for 2007, why do estimated not include all ages especially with dengue being more of a pediatric disease. In contrast to estimates in Buenos Aires at 682/100,000, the results of this study suggested a disease burden of 9,050 cases per 100,000 inhabitants, how reliable is the latter estimates and what is the explanation for the difference. According to the authors, the disparity between reported cases and disease burden as measured by seroprevalence indicates that many dengue infections are asymptomatic, and the cases reported through surveillance systems represent only a small fraction of the actual cases. The reviewer agrees that the majority of DENV infections is asymptomatic, then using seroprevalence data is not a reliable estimate of the burden of disease, whereas cases reported by passive surveillance will detect only the more severe cases, while an active surveillance will provide the best estimate of the number of cases, thus the best estimate of the burden of disease. Therefore, since the outcome of DENV infection cannot be determined based seroprevalence estimates, the reviewer suggest that an explanation be provided regarding their statement that serial populationbased seroprevalence studies can serve as a robust system for assessing disease burden. Overall, the strengths of this manuscript reflect efforts to understand the epidemiology of the rapidly growing and geographically expanding most important mosquito borne viral disease in the world, and the main weakness is the lack of a detailed description of the contents of the manuscript.

Specific comments:

The grammar need to be corrected mainly to for the use of past tense, the following are examples:

Lines 36 -37. “The overall response rate was 26.8%. Weighted estimates show a total of 193,707 people aged 18 and over have antibodies to DENV” (the “word” show should be “showed”).

Lines 149-151. The 149 performance parameters of this test, as reported by the manufacturer, indicate (the word” indicate” should be “indicated”) a sensitivity of 95.8%, 95% CI: 91.1-98.4%, and a specificity of 96.1%, 95% CI: 92.6- 151 98.4%, compared to ELISA assays.

Lines 210-211. In contrast, our study suggests (should be suggested) a disease burden of 9,050 cases 211 per 100,000 inhabitants

Lines 211-214. The disparity between reported cases and disease burden as measured by seroprevalence indicates (should be indicated) that many dengue infections are asymptomatic, and the cases reported through surveillance systems represent (should be represented ) only a small fraction of the actual cases

Lines 226-227. These findings are (replace “are” with “were”) consistent with those reported in the literature

Lines 234-236 According to data from the latest epidemiological bulletin, 235 68% of cases are (replace “are” with “were”) from DEN-2, while 32% are from DEN-1, suggesting that among 236 seropositive citizens there is homotypic immunity to both viruses, especially for DEN-2.

Lines 24 – 26. To estimate the seroprevalence of antibodies against the dengue virus (DENV) in adult residents of Buenos Aires, at the end of the 2023 season and immediately before the outbreak at the beginning of 2024. What caused the outbreaks and a brief statement of the health impact?

Lines 30-33 The sentences “The survey assessed the presence of IgG and IgM antibodies using a rapid chromatographic immunoassay. In positive cases, participants were invited for result confirmation through antibody determination using an enzyme-linked fluorescence assay (ELFA) technique” could be improved by considering the following: The survey determined the seroprevalence rate of DENV IgG and IgM antibodies using a rapid chromatographic immunoassay. Antibody positive participants were invited for re-testing to confirm antibody result using an enzyme-linked fluorescence assay (ELFA) technique.

Lines 42-44 states that “Seroprevalence results in adults in Buenos Aires show moderate exposure to the dengue virus (DENV), although still far from the thresholds usually considered for initiating a mass vaccination campaign”, suggest moderate “DENV infection rate” to replace the word, ” exposure” and delete dengue and use only DENV.

Lines 94-95 suggest that this sentence “Buenos Aires is the largest city in Argentina and has a population of 3.1 million of inhabitants, with 2.38 million having 18 or more years” be slightly revised to read with 2.38 million people 18 or more years of age.

In Table 1, the numerical values should be presented with commons rather than periods, for example 2.192.681 should be 2, 192,681, and for 100,0 should be 100 and for 193.707 should be 193,707/ Characteristics Age groups 18-34 years 595.269 should be 595,269, and for 27,1 should be 27.1 etc

Reviewer #2: The manuscript by Biscayart et al. describes the findings of the first dengue serosurvey conducted in Argentina. The study reveals a seroprevalence of 8%, with higher prevalence among individuals residing in slums, females, and those aged 18-34 years. The authors argue against dengue vaccination given the low seroprevalence in the population. While the manuscript presents valuable findings, I have the following comments and suggestions for improvement:

1. Selection of Primary Sampling Units (PSUs): The methodology for selecting PSUs from private households is detailed. However, the manuscript also references the inclusion of the slum population from the 'Informal Popular Neighbourhoods' sampling frame. The method for selecting PSUs from this frame is unclear. The authors should provide a description of how PSUs were selected for this population.

2. Sample size for the survey: 2. (First paragraph, page 5) Authors selected 3000 dwellings were selected. As per table 1, 2386388 individuals were tested for dengue antibodies. Is this correct? How much is the household/dwelling size in the city. It seems authors applied the seroprevalence estimated from the survey to the entire population of city. I feel Table 1 should only include the number of individuals actually tested and positive and not extrapolation to the entire population.

3. Clarification on Calibration by Fixed Margins Methodology: The manuscript refers to using "calibration by fixed margins," following the method developed by Deville and Särndal. As this method might not be familiar to all readers, it would be helpful if the authors could elaborate on how this calibration method was applied. Why did the authors choose this approach over using design weights and sampling methods, as typically recommended in Demographic and Health Surveys (DHS)?

4. Assumptions for Sample Size Estimation: Pl clarify how much was the anticipated prevalence while estimating the sample size.

5. Power Calculation: The manuscript mentions a power of 90% used in the sample size calculation. However, typical sample size formulas focus on the significance level (α) rather than statistical power. Could the authors clarify how power was considered in the calculation?

6. Adjustment for Assay Sensitivity and Specificity: The dengue seroprevalence was determined using a diagnostic test with a sensitivity of 95.8% and specificity of 96.1%. Given that the specificity is less than 100%, it would be prudent to adjust the observed seroprevalence to account for assay characteristics. Based on these values, the adjusted prevalence would be approximately 5.3%. This adjustment should be incorporated into the results.

7. Low Participation Rate and Generalizability: The participation rate in the study was notably low, with only 26% of the sampled population participating. This raises concerns about the generalizability of the findings. Can the authors provide any data to demonstrate that the characteristics of those who participated in the study were not significantly different from those who refused, particularly with respect to factors associated with dengue seropositivity?

8. Early Termination of Participant Recruitment: The manuscript mentions that participant recruitment had to be stopped earlier than planned (Line 271). Could the authors provide additional details on why recruitment was halted prematurely?

9. Inconsistency in Number of Positive Cases: On page 8, it is stated that 193,707 people tested positive for dengue (Line 173-177). However, earlier in the manuscript (Line 170), it is mentioned that only 804 interviews were conducted. This inconsistency needs clarification. Were the numbers based on extrapolation or actual testing?

10. Confirmatory Testing: How many of those who tested positive by the rapid test underwent confirmatory testing? This information should be included in the results section. Furthermore, the discussion mentions that only a small proportion confirmed their positive status. It would be helpful to quantify this.

11.Sensitivity and Specificity of Confirmatory Test: What were the sensitivity and specificity of the confirmatory test used in this study? Additionally, could the authors explain the rationale for conducting confirmatory testing?

12. Presentation of Factors Associated with Seropositivity: It would be beneficial to present the factors associated with dengue seropositivity in a table format for clearer interpretation. Interestingly, the confidence intervals for the odds ratios (ORs) appear to be very narrow. Did the authors compare the characteristics of those who tested positive and negative directly, or were these associations derived from the entire population based on extrapolated data?

PLOS authors have the option to publish the peer review history of their article (what does this mean? ). If published, this will include your full peer review and any attached files.

**Do you want your identity to be public for this peer review?** For information about this choice, including consent withdrawal, please see our Privacy Policy .

Reviewer #1: No

Reviewer #2: **Yes: ** Manoj Murhekar

---

## [Decision Letter · Decision Letter 1]

19 Jan 2025

PNTD-D-24-01249R1

Prevalence of Antibodies Against Dengue Virus in the City of Buenos Aires: Results of a Probabilistic Population Survey.

Dear Dr. Macchia,

Thank you for submitting your manuscript to PLOS Neglected Tropical Diseases. After careful consideration, we feel that it has merit but does not fully meet PLOS Neglected Tropical Diseases's publication criteria as it currently stands. Therefore, we invite you to submit a revised version of the manuscript that addresses the points raised during the review process.

Please submit your revised manuscript within 60 days Feb 18 2025 11:59PM. If you will need more time than this to complete your revisions, please reply to this message or contact the journal office at plosntds@plos.org. Please include the following items when submitting your revised manuscript:

We look forward to receiving your revised manuscript.

Kind regards,

Husain Poonawala

Academic Editor

David Safronetz

Section Editor

Shaden Kamhawi

co-Editor-in-Chief

Paul Brindley

co-Editor-in-Chief

**Reviewers' Comments:**

Reviewer's Responses to Questions

**Key Review Criteria Required for Acceptance?**

**Methods**

-Are the objectives of the study clearly articulated with a clear testable hypothesis stated?

-Is the study design appropriate to address the stated objectives?

-Is the population clearly described and appropriate for the hypothesis being tested?

-Is the sample size sufficient to ensure adequate power to address the hypothesis being tested?

-Were correct statistical analysis used to support conclusions?

-Are there concerns about ethical or regulatory requirements being met?

Reviewer #1: 1) While the rationale for this study is justified by efforts to better understand the epidemiology of dengue in Argentina, the reviewer’s comment to the authors to provide a brief description of the burden of this disease is not addressed in the Introduction section, and therefore, the author’s implied statement (line 24) that dengue is a rapidly growing health issue in Argentina based only on outbreaks does not support dengue as being a health issue. Why estimates of the disease burden is not presented is not understood, but would help to justify the rationale for conducting the study. That estimates were available is supported by the information presented in the Discussion section, lines 343 -345 that data from the latest epidemiological bulletin indicated that 68% of cases were from DEN-2, while 32% were caused by DEN-1. Also according to lines 293-297, “In Buenos Aires, using reported case data for those aged ≥18 years, the number of affected individuals was 16,138,294 (91% serologically confirmed) as of November 2023, generating an estimated prevalence of approximately 682 cases per 100,000 inhabitants. As stated, inclusion of this information in the Introduction would help to justify the rationale for conducting this study and for supporting the statement that dengue is a rapidly growing health issue in Argentina. Also, of relevance to including information of the disease burden, in lines 103-105, it is stated that “However, recommendations for implementing a vaccination program must consider many factors, among which the burden of infection is one of the most important. Since the majority of dengue infection are asymptomatic or mild, again, this should read the burden of disease and not the burden of infection.

2) While the implied purpose of serosurvey in this study was to determine if a dengue vaccination campaign was needed, the specific purpose of the study is vague, for example, it is stated in lines 112-116 that “Considering the heterogeneity in the distribution of dengue disease both between and within countries, the WHO Strategic Advisory Group of Experts (SAGE) recommends the implementation of local epidemiology plans (5). In this manuscript, we report the first determination of dengue seroprevalence in Argentina, particularly in the City of Buenos Aires, using a probabilistic approach”. Does this WHO plan recommend this seroprevalence study, and if so, it is not clear how this study will contribute to the understanding of dengue disease.

The study site could be improved by more information on the targeted population, for example, a breakdown of the total population of the slum section versus other socially and demographic sectors of the city, such as occupations, water use practices, sanitation,, health care, and mosquito control services. As described, the information provided is that general private households’ frame, comprises approximately 93% of the dwellings in Buenos Aires, while the second frame, called the “Informal Popular Neighborhoods” frame, that included the dwellings in the city’s slums

3) The comment by the reviewer as to why only adults 18 years of age and older were enrolled in this study was addressed by the authors (lines 391-401), but the reason should be discussed regarding the potential impact of the exclusion of the younger ages on the validity of the results as being representative of the entire population.as stated by the authors that the results may have limited generalizability to younger age groups, and therefore interpreted by the reviewer as posing the critical question regarding the repeated claim that the finding were representative of the entire population. According to the WHO’s policies, estimates of infection/disease burden for making a decision on vaccination campaigns should be based on the results of studies involving school-aged children, 5-18 years of age.

4) Line 412 The design of the sampling method did not appear consider the well documented observation that dengue virus transmission is known to occur in spatial clusters. On the basis of this observation and the very low number of samples and especially the low seroprevalence, especially by the limited number per age groups need to be addressed regarding the validity of the representation of the total population and addressed as a possible limitation of the outcome of the study.

5) The reviewer suggest that in addition to the city and country name of the assay manufacturers, and if available, publication(s) in support of the sensitivity and specificity of the assay data be added to the manuscript. Also, include the dilution made automatically and the reason for collecting blood samples a second time. Was there agreement of the results of all samples between the 2 assays. Were the manufacturer’s data on sensitivity and specificity validated and if so, describe.

**Results**

-Does the analysis presented match the analysis plan?

-Are the results clearly and completely presented?

-Are the figures (Tables, Images) of sufficient quality for clarity?

Reviewer #1: 1) Since the outcome of DENV infection cannot be determined based seroprevalence estimates, the reviewer suggest that an explanation be provided regarding the statement that serial population based seroprevalence studies can serve as a robust system for assessing disease burden.

2) According to the authors, the disparity between the reported symptomatic cases of 682 per 100,000 reported based on assumed passive surveillance as of November 2023 in Buenos Aires and the estimate 9,050 cases per 100,000 observed in this study was because many of the dengue virus infections detected in the seroprevalence study were asymptomatic. While many dengue virus infections are asymptomatic, how was estimate of symptomatic cases based on seroprevalence data determined?

3) Lines 319-325 According to the results, “The age distribution of DENV antibody-positive participants showed a higher prevalence in younger adults as compared to the older age groups”. As implied by the authors, this observation is not consistent with the existing understanding of the epidemiology of dengue even in non-hyper-endemic areas, and it is not understood how according to the authors, the concentration of more recent outbreaks could occur among the younger, more mobile populations. Furthermore, it is not understood how the observation “underscores the need for targeted prevention strategies and ongoing surveillance to monitor shifting transmission dynamics”.) Lines 26-28

4)As a major weakness of this study that is not mentioned in the results section, but is presented in the discussion section that participant recruitment had to be stopped earlier than planned. According to the authors, recruitment did not begin until September, 2023 and ended immediately before the subsequent dengue epidemic in early 2024, the specific date of the survey being from September 21 until December 1, 2023.The reason according to the authors was because of the unusually aggressive dengue outbreak recorded across Argentina since December 2023, which is not understood, is this year correct? Considering that recruitment began in September 2023, the collected data reflected the prevalence status before or during the beginning of the 2024 new epidemic. According to the authors, “Under these conditions, it was decided to stop recruitment and take an estimate of dengue seroprevalence before the epidemic in early 2024 in Argentina”. The reviewer’s is led to believe that the enrollment of only 808 of the planned number of 2,998 participants was because of the very low enrollment rate, but then according to the authors, the reason was because of the study was aborted because of the dengue epidemic. Also, with the largest ongoing outbreak ever recorded in Argentina, the authors need to provide an explanation for the very low dengue IgG and IgM antibody prevalent rate, especially only a 1.5% IgM antibody rate that would not appear to adequately reflect the recent infection or during the past 2 or more months the estimated duration of IgM antibody.

5) In the discussion section it is stated that “Although dengue is a notifiable disease, it is likely that the number of reported cases underestimates the disease burden due to the presence of asymptomatic or mildly symptomatic cases, as well as a tendency for underreporting communicable diseases”. It is not clear as to the meaning of the latter sentence, as asymptomatic cases would not contribute to the burden of disease, under-reporting would lessen the estimated burden of disease, but is there evidence for under-reporting of cases.

**Conclusions**

-Are the conclusions supported by the data presented?

-Are the limitations of analysis clearly described?

-Do the authors discuss how these data can be helpful to advance our understanding of the topic under study?

-Is public health relevance addressed?

Reviewer #1: (No Response)

**Editorial and Data Presentation Modifications?**

Reviewer #1: (No Response)

**Summary and General Comments**

Reviewer #1: his revised manuscript describes the author’s study to determine the prevalence of dengue virus (DENV) antibody in adult residents after the largest recorded outbreak of dengue during the 2023 and immediately before the beginning of a subsequent dengue outbreak in early 2024 in Buenos Aires, Argentina. While the authors address some but not all of the reviewer’s comments. As stated by the authors, lines 379-393, “Our study has strengths and weaknesses that should be mentioned”. The following observations includes the author’s limitations and add others that raise concern about the scientific rigor of the modified study plans and the validity of the observations generated by this study as follows:

1)Firstly, participant recruitment had to be stopped earlier than planned. The main reason is not clear, but understood to be because of the unusually aggressive dengue outbreak recorded across Argentina since December 2023. Considering that recruitment began in September 2023, the collected data reflected the prevalence status before the new epidemic. Under these conditions, it was decided to stop recruitment and take an estimate of dengue seroprevalence before the most recent epidemic in early 2024 in Argentina. Ongoing surveys will provide information on how this situation may have changed. As understood, surveillance was terminated December 1, 2023, what ongoing surveillance is being referred too?

2) Additionally, the non-response rate was higher than expected. As stated in lines 233- 234 and lines 246- 248, “Out of the 2,998 selected households, 804 individuals participated in the study, resulting in a response rate of only 26.8%”. The low participation rate introduces the possibility of non-response bias. Although calibration by fixed margins was applied to mitigate this issue, the lack of data on non-participants limits our ability to directly assess differences between participants and those who refused to participate. The very low enrollment of 804 participants resulted in only about 200 participants per age group with no assurance that these participants were representative of the entire population of 2.38 million people.

3) The proportion of participants who underwent confirmatory testing was very limited (27/71), which may affect the generalizability of the confirmatory results. However, the 100% concordance observed supports the reliability of the rapid test in this context. With the interruption of the study plans, especially the substantial reduction in the number of enrolled participants, it is not clear as to how reliable the estimated prevalence rate was for the entire city and the observations to confirm or refute the validity of the extent of being representative are not presented in this study .

4) The exclusion of the under 18 years of age in the study is likely to have resulted in a bias that is not accounted for regarding the overall prevalence of dengue virus antibody in the entire population of the city. An explanation is provided for the exclusion of under the age of 18 year, and while stating that the results may have limited generalizability to younger age groups, and therefore raises critical question regarding the repeated claim that the finding were representative of the entire population.

5) The validity of using an approach based on weighted estimates for dengue virus seroprevalence could be biased because this approach would assumed that the risk of dengue infection is homogeneous, an assumption that is not clearly addressed in this study, for example variation in the distribution of Aedes aegypti, that is a major determinant of temporal and spatial distribution of dengue virus infection, and that the slum population differed from the other 93% of the population is not supported by any relevant information in regards to the potential risk of dengue virus infection.

6) With the largest ongoing outbreak ever recorded in Argentina during 2023, what is the explanation for the very low dengue IgG antibody (8%) and IgM antibody prevalent rate, especially the 1.5% IgM rate disease, a rate that indicate a low risk for DENV infection. Also, the conclusions that the results of this study showed a modest burden of dengue in the adult population of Buenos Aires immediately before the 2024 outbreak is not supported as seroprevalence cannot be used to estimate the burden of dengue disease.

Overall, the strengths of this manuscript reflect efforts to understand the epidemiology of the rapidly growing and geographically expanding most important mosquito borne viral disease in the world, and the main weakness is the lack of clarity, failure to implement the described plan, and the critical question of the reliability of the findings raising the question of the validity of the conclusion that a modest burden of dengue existed in the adult population of Buenos Aires.

Specific comments

Aedes aegypti and Aedes albopictus need to be Italicized throughout the manuscript.

In lines 21 -23 it is stated that, “The VIDAS Anti-Dengue IgG assay was selected for confirmation due to its higher specificity (99.1%) and semi-quantitative design, which complements the qualitative nature of the rapid test”. As such, Anti-Dengue IgG should read Dengue IgG antibody as Anti is redundant and should include antibody to be more specific to replace Anti

In lines 126-132, as examples of the misuse of grammar as well as elsewhere in the manuscript, the past tense should be used as follows: “Consequently, the sample results (add were and delete are) valid representations of the entire city. This approach (add allowed and delete allows) for the inclusion of diverse subpopulations by dividing the sampling frame into strata based on relevant criteria, such as income levels and geographic areas. By employing multi-stage sampling, we (add optimized and delete optimize) resource allocation while maintaining statistical reliability, as each stage progressively (add narrowed and delete narrows) the selection from broader units (e.g., census tracts) to individual households and participants. This methodology (add reduced and delete reduce) sampling bias and (add enhanced and delete enhances) enhances the validity of results for an urban area as complex and diverse as Buenos Aires.

In lines 403-405 it is stated that “Most DENV infections are asymptomatic, and antibody detection reflects prior exposure rather than current infection rates”. Suggest this sentence be revised to read, “Most DENV infections are asymptomatic, and antibody detection reflects prior infections rather than current infection rates”.

PLOS authors have the option to publish the peer review history of their article (what does this mean? ). If published, this will include your full peer review and any attached files.

**Do you want your identity to be public for this peer review?** For information about this choice, including consent withdrawal, please see our Privacy Policy .

Reviewer #1: No

**Figure resubmission:**
---

## [Editor Report · Decision Letter 2]

4 Feb 2025

Dear Dr. Macchia,

We are pleased to inform you that your manuscript 'Prevalence of Antibodies Against Dengue Virus in the City of Buenos Aires: Results of a Probabilistic Population Survey.' has been provisionally accepted for publication in PLOS Neglected Tropical Diseases.

Best regards,

Husain Poonawala

Academic Editor

David Safronetz

Section Editor

Shaden Kamhawi

co-Editor-in-Chief

Paul Brindley

co-Editor-in-Chief

---

## [Editor Report · Acceptance letter]

Dear Dr. Macchia,

We are delighted to inform you that your manuscript, "Prevalence of Antibodies Against Dengue Virus in the City of Buenos Aires: Results of a Probabilistic Population Survey.," has been formally accepted for publication in PLOS Neglected Tropical Diseases.

Best regards,

Shaden Kamhawi

co-Editor-in-Chief

Paul Brindley

co-Editor-in-Chief
